# Algebraic structure of classical integrability for complex sine-Gordon

**Jean Avan[1], Luc Frappat[2★] and Eric Ragoucy[2]**

**1** Laboratoire de Physique Théorique et Modélisation,
CY Cergy Paris Université, CNRS, F-95302 Cergy-Pontoise, France
**2** Laboratoire d'Annecy-le-Vieux de Physique Théorique LAPTh,
Université Grenoble Alpes, USMB, CNRS, F-74000 Annecy

★ luc.frappat@lapth.cnrs.fr

## Abstract

The algebraic structure underlying the classical $r$-matrix formulation of the complex sine-Gordon model is fully elucidated. It is characterized by two matrices $a$ and $s$, components of the $r$ matrix as $r = a - s$. They obey a modified classical reflection/Yang–Baxter set of equations, further deformed by non-abelian dynamical shift terms along the dual Lie algebra $su(2)^*$. The sign shift pattern of this deformation has the signature of the twisted boundary dynamical algebra. Issues related to the quantization of this algebraic structure and the formulation of quantum complex sine-Gordon on those lines are introduced and discussed.

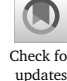 Check for updates

## 1 Introduction

The complex sine-Gordon (CSG) field theory is a two-dimensional relativistic model with $U(1)$ symmetry, exhibiting classical and quantum integrability features. It was originally derived by Pohlmeyer [1] from the $O(4)$ linear sigma model. Lund and Regge [2] independently constructed it in the context of condensed matter physics, and proposed a Lagrangian formulation and a Lax pair description. An alternative Lagrangian using a different set of fields was proposed by Getmanov [3] and the corresponding Lax pair was derived by De Vega and Maillet [4]. A later interpretation of the CSG model as a coset $SU(2)/U(1)$ WZW model perturbed by thermal operator was derived by Bakas [5].

Some aspects of classical integrability were discussed in the earliest days of the model, with the construction of multi-soliton solutions by classical inverse scattering method applied to the Lax pair [2, 3]. Quantum integrability was approached essentially by perturbative methods, originating with the works of De Vega and Maillet [4, 6], and later extended by Dorey and Hollowood [7]. It was noted at the time that quantum integrability breaks down at one-loop level and requires addition of counterterms to the Lagrangian. This situation was understood in the context of the WZW formulation of the CSG model.

We wish to emphasize that most of these derivations were undertaken in the Lagrangian framework and, as we already indicated, by semi-classical or perturbative methods. To the best of our knowledge, an Hamiltonian treatment at the quantum level is still missing. One lacks in particular an algebraic formulation of the $r$-matrix structure for the classical Lax matrix, useful to define a quantum model preserving at least some features of integrability through its derived algebraic structure. It may help understanding the integrability breaking mechanism and its subsequent restoring. This approach constitutes in any case a fundamentally different way of tackling the particular features of quantum CSG with respect to its integrability properties. A starting point to achieve such a quantum formulation is to completely unravel the classical Hamiltonian formulation by $r$-matrix formalism proposed by Maillet [8]. Moreover, this known $r$-matrix exhibits characteristic features of a dynamical structure, albeit not of Gervais–Neveu–Felder type (*abelian* dynamical structure) [9, 10]. This renders its elucidation interesting for its own sake. We identify this dynamical structure in the present article: the classical $r$-matrix structure of the CSG model is a *modified*, *non-abelian SU(2)* boundary-dynamical, classical Yang–Baxter reflection equation for an $(a, s)$ pair, where $r = a - s$. The terminology used here will be explained in detail in the next section. Section 3 is devoted to several issues regarding the quantization of the unravelled algebraic structure.

## 2   The classical integrability structure

Let us first prepare the notations. We adopt the field variables $(\psi, \bar{\psi})$ in two-dimensional space-time $(x, t)$ defined in [3] and the corresponding Lax formulation in [4, 8]. The Lagrangian of the classical CSG is defined by:

$$\mathscr{L} = \frac{1}{2g}\left(\frac{\partial_\mu \psi \, \partial^\mu \bar{\psi}}{1 - \psi\bar{\psi}} - m^2 \psi\bar{\psi}\right), \tag{1}$$

where $\psi(x, t)$ is a complex field, $\bar{\psi}(x, t)$ its complex conjugate, $g$ the coupling constant, and $m$ the mass ($g$ and $m$ are set to 1 in the following). The conjugated momenta $\pi$ and $\bar{\pi}$ to the fields $\psi$ and $\bar{\psi}$ are given by

$$\pi = \frac{\partial_t \bar{\psi}}{2(1 - \psi\bar{\psi})} \qquad \text{and} \qquad \bar{\pi} = \frac{\partial_t \psi}{2(1 - \psi\bar{\psi})}. \tag{2}$$

The corresponding (equal-time) canonical Poisson structure is therefore

$$\{\pi(x, t), \psi(y, t)\} = \delta(x - y) \qquad \text{and} \qquad \{\bar{\pi}(x, t), \bar{\psi}(y, t)\} = \delta(x - y). \tag{3}$$

The associated Lax pair is given by [8]:

$$\begin{aligned}
L(x, u) = \frac{i}{4}\bigg(&\left(-u + \frac{1 - \psi\bar{\psi}}{u} + 2i(\psi\pi - \bar{\psi}\bar{\pi})\right)\sigma^z \\
&+ 2i\left(2\sqrt{1 - \psi\bar{\psi}}\,\bar{\pi} + \frac{\partial_x \psi}{\sqrt{1 - \psi\bar{\psi}}} - \frac{i}{u}\sqrt{1 - \psi\bar{\psi}}\,\psi\right)\sigma^+ \\
&- 2i\left(2\sqrt{1 - \psi\bar{\psi}}\,\pi + \frac{\partial_x \bar{\psi}}{\sqrt{1 - \psi\bar{\psi}}} + \frac{i}{u}\sqrt{1 - \psi\bar{\psi}}\,\bar{\psi}\right)\sigma^-\bigg)
\end{aligned} \tag{4}$$

and

$$
\begin{aligned}
M(x,u) = \frac{i}{4}\Bigg(\Big(&-u-\frac{1-\psi\bar\psi}{u}+i\,\frac{\psi\partial_x\bar\psi-\bar\psi\partial_x\psi}{1-\psi\bar\psi}\Big)\sigma^z \\
&+2i\Big(2\sqrt{1-\psi\bar\psi}\,\bar\pi+\frac{\partial_x\psi}{\sqrt{1-\psi\bar\psi}}+\frac{i}{u}\sqrt{1-\psi\bar\psi}\,\psi\Big)\sigma^+ \\
&-2i\Big(2\sqrt{1-\psi\bar\psi}\,\pi+\frac{\partial_x\bar\psi}{\sqrt{1-\psi\bar\psi}}-\frac{i}{u}\sqrt{1-\psi\bar\psi}\,\bar\psi\Big)\sigma^-\Bigg),
\end{aligned}
\tag{5}
$$

where $\sigma^\pm,\sigma^z$ are the usual Pauli matrices satisfying $[\sigma^z,\sigma^\pm]=\pm2\sigma^\pm$ and $[\sigma^+,\sigma^-]=\sigma^z$. The Lax pair matrices $L$ and $M$ depend on the $x$ variable only through the fields $\psi$, $\bar\psi$ and their derivatives.

The ultralocal Poisson structure (3) endows the Lax matrix $L$ with a non-ultralocal $r$-matrix Poisson structure given by

$$
\begin{aligned}
\{L(x,u_1)\overset{\otimes}{,}L(y,u_2)\}=\Big(&\partial_x a(x,u_1,u_2)+\big[a(x,u_1,u_2)-s(x,u_1,u_2),L(x,u_1)\otimes 1\big] \\
&+\big[a(x,u_1,u_2)+s(x,u_1,u_2),1\otimes L(x,u_2)\big]\Big)\delta(x-y) \\
&+\tfrac{1}{2}\big(s(x,u_1,u_2)+s(y,u_1,u_2)\big)(\partial_x-\partial_y)\delta(x-y).
\end{aligned}
\tag{6}
$$

The matrices $a(x,u_1,u_2)$ and $s(x,u_1,u_2)$ read

$$
a(x,u_1,u_2)=-\tfrac{1}{2}P\,\frac{u_1+u_2}{u_1-u_2}+\frac{1}{8\sqrt{1-\psi\bar\psi}}\big((\psi\sigma^++\bar\psi\sigma^-)\otimes\sigma^z-\sigma^z\otimes(\psi\sigma^++\bar\psi\sigma^-)\big),
$$

$$
\begin{aligned}
s(x,u_1,u_2)=&\frac{1}{8\sqrt{1-\psi\bar\psi}}\big((\psi\sigma^++\bar\psi\sigma^-)\otimes\sigma^z+\sigma^z\otimes(\psi\sigma^++\bar\psi\sigma^-)\big) \\
&-\tfrac{1}{2}(\sigma^+\otimes\sigma^-+\sigma^-\otimes\sigma^+),
\end{aligned}
\tag{7}
$$

where $P$ is the usual permutation matrix. The matrices $a$ and $s$ are respectively antisymmetric and symmetric under the exchange of the tensor product, $Pa(x,u_1,u_2)P=-a(x,u_2,u_1)$ and $Ps(x,u_1,u_2)P=s(x,u_2,u_1)$, guaranteeing the antisymmetry of the Poisson structure (6).

Defining the $r$-matrix as $r(x,u_1,u_2)=a(x,u_1,u_2)-s(x,u_1,u_2)$, it can be checked that it satisfies the following dynamical classical Yang–Baxter equation [8]:

$$
\begin{aligned}
\big[r_{12}(x,u_1,u_2),r_{13}(x,u_1,u_3)\big]&+\big[r_{12}(x,u_1,u_2),r_{23}(x,u_2,u_3)\big]+\big[r_{32}(x,u_3,u_2),r_{13}(x,u_1,u_3)\big] \\
&+K_{123}(x,u_1,u_2,u_3)-K_{132}(x,u_1,u_3,u_2)=0.
\end{aligned}
\tag{8}
$$

As usual, this equation lies in a three-fold tensor product, the indices indicating in which spaces the matrices act non trivially. The kernel $K_{ijk}(x,u_i,u_j,u_k)$ is defined by

$$
\{r_{ij}(x,u_i,u_j),L_k(y,u_k)\}=K_{ijk}(x,u_i,u_j,u_k)\,\delta(x-y).
\tag{9}
$$

We first establish an explicit algebraic expression for this Poisson bracket. We introduce the differential operators:

$$
J^z=2\left(\bar\psi\,\frac{\partial}{\partial\bar\psi}-\psi\,\frac{\partial}{\partial\psi}\right),\quad J^+=2\sqrt{1-\psi\bar\psi}\,\frac{\partial}{\partial\psi},\quad J^-=-2\sqrt{1-\psi\bar\psi}\,\frac{\partial}{\partial\bar\psi}.
\tag{10}
$$

They satisfy the commutation relations of the $sl(2)$ algebra:

$$
\big[J^z,J^\pm\big]=\pm2J^\pm,\quad\big[J^+,J^-\big]=J^z.
\tag{11}
$$

The Poisson bracket of the matrix $r$ with the Lax matrix $L$ now takes an algebraic form:

**Proposition 2.1** *The kernel $K_{123}(x, u_1, u_2, u_3)$ is given by*

$$K_{123}(x, u_1, u_2, u_3) = -2 \sum_{a,b=z,\pm} K_{ab}^{-1} J^a \, r_{12}(x, u_1, u_2) \otimes \sigma^b \,, \tag{12}$$

*where $K_{ab}$ is the Killing form of $su(2)$.*

**Proof:** By definition of the Poisson bracket, one has

$$\{r_{12}(x, u_1, u_2), L_3(y, u_3)\} = \frac{\partial r_{12}}{\partial \pi} \frac{\partial L_3}{\partial \psi} + \frac{\partial r_{12}}{\partial \bar{\pi}} \frac{\partial L_3}{\partial \bar{\psi}} - \left( \frac{\partial r_{12}}{\partial \psi} \frac{\partial L_3}{\partial \pi} + \frac{\partial r_{12}}{\partial \bar{\psi}} \frac{\partial L_3}{\partial \bar{\pi}} \right), \tag{13}$$

that is

$$K_{123}(x, u_1, u_2, u_3) = \frac{1}{2} \frac{\partial}{\partial \psi} \, r_{12}(x, u_1, u_2) \otimes \left( \psi \sigma^z - 2\sqrt{1 - \psi\bar{\psi}} \, \sigma^- \right)$$

$$- \frac{1}{2} \frac{\partial}{\partial \bar{\psi}} \, r_{12}(x, u_1, u_2) \otimes \left( \bar{\psi} \sigma^z - 2\sqrt{1 - \psi\bar{\psi}} \, \sigma^+ \right). \tag{14}$$

It follows that (14) can be written as (12). ∎

We now establish the precise meaning of (8) by deriving explicit Yang–Baxter type equations for the separate matrices $a$ and $s$. They exhibit the same pattern, albeit with a now explicit algebraic form for the supplementary Poisson bracket terms obtained from Proposition 2.1. This is the second main result of this paper.

**Proposition 2.2** *The matrices $a$ and $s$ satisfy the following modified classical Yang–Baxter equations (MCYBE):*

$$\left[ a_{12}(x, u_1, u_2), a_{13}(x, u_1, u_3) \right] + \left[ a_{12}(x, u_1, u_2), a_{23}(x, u_2, u_3) \right] + \left[ a_{13}(x, u_1, u_3), a_{23}(x, u_2, u_3) \right]$$

$$+ \tfrac{1}{2} \left( K_{123}^{(a)}(x, u_1, u_2, u_3) - K_{132}^{(a)}(x, u_1, u_3, u_2) + K_{231}^{(a)}(x, u_2, u_3, u_1) \right) = -\Omega_{123} \,, \tag{15}$$

$$\left[ a_{12}(x, u_1, u_2), s_{13}(x, u_1, u_3) \right] + \left[ a_{12}(x, u_1, u_2), s_{23}(x, u_2, u_3) \right] + \left[ s_{13}(x, u_1, u_3), s_{23}(x, u_2, u_3) \right]$$

$$+ \tfrac{1}{2} \left( -K_{123}^{(a)}(x, u_1, u_2, u_3) - K_{132}^{(s)}(x, u_1, u_3, u_2) + K_{231}^{(s)}(x, u_2, u_3, u_1) \right) = -\Omega_{123} \,, \tag{16}$$

*where*

$$\Omega_{123} = \tfrac{1}{8} \sum_{\tau \in \mathfrak{S}_3} \varepsilon(\tau) \, \sigma^{\tau(z)} \otimes \sigma^{\tau(+)} \otimes \sigma^{\tau(-)}. \tag{17}$$

*$\mathfrak{S}_3$ is the permutation group of three elements and $\varepsilon(\tau)$ the signature of the permutation $\tau$. The kernels $K_{ijk}^{(a)}(x, u_i, u_j, u_k)$ and $K_{ijk}^{(s)}(x, u_i, u_j, u_k)$ are defined as in (9), and expressed as in (12) for the matrices $a$ and $s$. Their explicit form is given in (18) and (19).*

**Proof:** Let us start with (8) with $r(x, u_1, u_2) = a(x, u_1, u_2) - s(x, u_1, u_2)$. A straightforward calculation shows that the commutator part of (8) is expressed as $Y_{123}^{(a)} - Y_{123}^{(s)} + Y_{132}^{(s)} + Y_{231}^{(s)}$ where

$$Y_{123}^{(a)} = \left[ a_{12}(x, u_1, u_2), a_{13}(x, u_1, u_3) \right] + \left[ a_{12}(x, u_1, u_2), a_{23}(x, u_2, u_3) \right]$$

$$+ \left[ a_{13}(x, u_1, u_3), a_{23}(x, u_2, u_3) \right]$$

and

$$Y_{123}^{(s)} = \left[ a_{12}(x, u_1, u_2), s_{13}(x, u_1, u_3) \right] + \left[ a_{12}(x, u_1, u_2), s_{23}(x, u_2, u_3) \right]$$

$$+ \left[ s_{13}(x, u_1, u_3), s_{23}(x, u_2, u_3) \right].$$

Hence a natural question is to understand whether CYBE type equations hold for the matrices $a$ and $s$, in other words to investigate how to complete $Y_{123}^{(a)}$ and $Y_{123}^{(s)}$ in order to obtain CYBE type equations, allowing to reproduce (8). Since $K_{123} = K_{123}^{(a)} - K_{123}^{(s)}$, it is useful to evaluate these last two kernels. From the expression (12) applied to the matrices $a$ and $s$, one has:

$$K_{123}^{(a)} = \frac{1}{16\sqrt{1-\psi\bar{\psi}}} \left( (\psi\sigma^+ - \bar{\psi}\sigma^-) \otimes \sigma^z \otimes \sigma^z - \sigma^z \otimes (\psi\sigma^+ - \bar{\psi}\sigma^-) \otimes \sigma^z \right)$$
$$+ \frac{1}{16(1-\psi\bar{\psi})} \left( (\psi\sigma^+ + \bar{\psi}\sigma^-) \otimes \sigma^z \otimes (\psi\sigma^+ - \bar{\psi}\sigma^-) - \sigma^z \otimes (\psi\sigma^+ + \bar{\psi}\sigma^-) \otimes (\psi\sigma^+ - \bar{\psi}\sigma^-) \right)$$
$$+ \frac{1}{8} \left( \sigma^z \otimes \sigma^+ \otimes \sigma^- - \sigma^z \otimes \sigma^- \otimes \sigma^+ - \sigma^+ \otimes \sigma^z \otimes \sigma^- + \sigma^- \otimes \sigma^z \otimes \sigma^+ \right) \tag{18}$$

and

$$K_{123}^{(s)} = \frac{1}{16\sqrt{1-\psi\bar{\psi}}} \left( (\psi\sigma^+ - \bar{\psi}\sigma^-) \otimes \sigma^z \otimes \sigma^z + \sigma^z \otimes (\psi\sigma^+ - \bar{\psi}\sigma^-) \otimes \sigma^z \right)$$
$$+ \frac{1}{16(1-\psi\bar{\psi})} \left( (\psi\sigma^+ + \bar{\psi}\sigma^-) \otimes \sigma^z \otimes (\psi\sigma^+ - \bar{\psi}\sigma^-) + \sigma^z \otimes (\psi\sigma^+ + \bar{\psi}\sigma^-) \otimes (\psi\sigma^+ - \bar{\psi}\sigma^-) \right)$$
$$+ \frac{1}{8} \left( \sigma^z \otimes \sigma^- \otimes \sigma^+ - \sigma^z \otimes \sigma^+ \otimes \sigma^- - \sigma^+ \otimes \sigma^z \otimes \sigma^- + \sigma^- \otimes \sigma^z \otimes \sigma^+ \right). \tag{19}$$

$Y_{123}^{(a)}$ and $Y_{123}^{(s)}$ do not contain any term in $\psi^2$ nor $\bar{\psi}^2$, hence the possible combinations liable to compensate $Y_{123}^{(a)}$ and $Y_{123}^{(s)}$ respectively, are $\frac{1}{2}(K_{123}^{(a)} - K_{132}^{(a)} + K_{231}^{(a)})$ and $\frac{1}{2}(-K_{123}^{(a)} - K_{132}^{(s)} + K_{231}^{(s)})$. A direct computation then leads to equations (15) and (16). It is interesting to note the necessary occurence of the term $\Omega_{123}$ in these equations, which cancels out when reproducing (8). It is a structural feature of the equations for the matrices $a$ and $s$. Its occurence now characterizes (15) and (16) as *modified* CYBE with a dynamical extension. ∎

The modified classical Yang–Baxter equation (15) (with or without its added dynamical shift) is a well-known object described in e.g. [11] (without dynamics) or [12, 13] (with dynamics). The adjoint modified classical Yang–Baxter set (16) however, to the best of our knowledge, has not been identified in a given system or defined a priori before. It will therefore need to be better understood, hopefully from the quantum point of view as a semi-classical limit of the full reflection algebra structure. We shall come back to this point in the next section.

It is immediate to check that the differential operators (10) realize a linear representation (spin 1 $su(2)$), when acting on the three functions $x^+ = \bar{\psi}$, $x^0 = 2\sqrt{1-\psi\bar{\psi}}$ and $x^- = \psi$. Since $x^\pm$ are complex conjugate, it indicates that the correct deformation parameter manifold is the sphere

$$x^+ x^- + \frac{1}{4}(x^0)^2 = 1. \tag{20}$$

The classical Poisson algebra is therefore identified with a non-abelian $su(2)^*$ dynamical reflection $(a, s)$ structure, however realized on a moduli space of deformations which is a submanifold (20) of the full dual space $su(2)^*$. This is directly identified with the proposed construction of CSG as a (deformed) WZWN model on $SU(2)/U(1)$ by Bakas [5]. The CSG model was recovered there from a $SU(2)$ WZWN model in a two-step procedure: first of all gauging a $U(1)$ subgroup, chosen to be the diagonal one. This yields the massless part of the CSG Lagrangian after a suitable gauge fixing, with the exact reduced field content $\psi$, $\bar{\psi}$, and gauge condition (20). The additional mass term is then identified as a perturbation of WZWN by the first thermal operator. Finally, the notion of CYBE with non-abelian dynamical shifts has been considered in [12], and further developed in [13] where a quantization procedure was proposed.

These three coordinates now allow to write $a$ and $s$ in a very compact way, as a projective canonical element. It only involves a canonical product of the *projective* components $(x^+, x^0, x^-)$ of the element in the dual Lie algebra $su(2)^*$ parametrizing the deformation, and the three Pauli matrices, basis of the Lie algebra $su(2)$, namely $\boldsymbol{x} \cdot \boldsymbol{\sigma} = x^+ \sigma^- + x^- \sigma^+ + \frac{1}{2} x^0 \sigma^z$:

$$a(u_1, u_2) = -\tfrac{1}{2} P \frac{u_1 + u_2}{u_1 - u_2} + \frac{1}{4x^0}\big(\boldsymbol{x} \cdot \boldsymbol{\sigma} \otimes \sigma^z - \sigma^z \otimes \boldsymbol{x} \cdot \boldsymbol{\sigma}\big), \tag{21}$$

$$s(u_1, u_2) = \tfrac{1}{4} \, \mathbb{I} \otimes \mathbb{I} - \tfrac{1}{2} P + \frac{1}{4x^0}\big(\boldsymbol{x} \cdot \boldsymbol{\sigma} \otimes \sigma^z + \sigma^z \otimes \boldsymbol{x} \cdot \boldsymbol{\sigma}\big). \tag{22}$$

A distinctive feature of (21) must be pointed out here. Classical non-abelian dynamical skew-symmetric $r$-matrices such as $a$ are usually assumed, in addition to DCYBE (15), to obey an equivariance property (extending the zero-weight condition in the abelian case, see [9,10]). Namely, viewed as functions $a : \mathfrak{h}^* \to H \otimes H$, the adjoint action of $\mathfrak{h}$ over $H \otimes H$ is identical to the coadjoint action of $\mathfrak{h}$ over $\mathfrak{h}^*$ [12,13]. It is clear that (21) cannot obey such a property. Indeed, the adjoint action of $\sigma^\pm$ in $su(2)$ over $a$ generates *new* terms such as $\sigma^\pm \otimes \sigma^\mp - \sigma^\mp \otimes \sigma^\pm$ in $a$, which cannot be compensated by $ad^* \sigma^\pm$ over $a$. This issue and its connection to associativity in the quantum case will be addressed presently.

# 3  Open problems about quantization

We have now established and fully characterized the classical $r$-matrix structure underlying integrability properties of the complex sine-Gordon model. We have identified a pair of anti-symmetric ($a$) and symmetric ($s$) matrices, combining as $r = a - s$ to yield the classical CSG $r$-matrix. They respectively obey a non-abelian $su(2)$ dynamical modified classical Yang–Baxter equation (for $a$), and its adjoint extension ($a$ acting on $s$).

In order to obtain a consistent formulation of the quantum integrability properties for CSG, we now need to first of all propose a quantum version of the $(a, s)$ matrix Poisson algebra derived above. We then need to construct a quantum Lax matrix reproducing the Poisson algebraic structure proposed in [8] and detailed in (6). These questions raise delicate issues which we now detail, and (for some of them) propose ideas of resolution. The full issue of quantum integrability for CSG is known to be not straightforward, since "classical" conserved quantities break down at one-loop level, but consistent implementations of the Lagrangian restore integrability [7]. Identifying obstructions to quantization of the classical integrability structure should help in characterizing a systematic procedure for restoration of integrability at the quantum level, hopefully non-perturbatively. This is our ultimate purpose here.

The starting point to quantize (15)–(16) is to recognize (without the supplementary term $\Omega_{123}$) the characteristic features of a dynamical reflection algebra of the type "alternative boundary dynamical" or "twisted dynamical" [14]. We recall that, in the abelian dynamical case, three dynamical extensions are known for the general [15] quadratic algebra $A_{12} K_1 B_{12} K_2 = K_2 C_{12} K_1 D_{12}$, parametrized as

$$A_{12} K_1(q - \varepsilon_R h^{(2)}) B_{12} K_2(q + \varepsilon_L h^{(1)}) = K_2(q - \varepsilon_R h^{(1)}) C_{12} K_1(q + \varepsilon_L h^{(2)}) D_{12}. \tag{23}$$

$\varepsilon_R = -\varepsilon_L$ defines the boundary dynamical algebra [16,17], $\varepsilon_R = \varepsilon_L$ defines the twisted boundary dynamical algebra [14], $\varepsilon_R = 0, \varepsilon_L = 1$ defines semi-dynamical algebras [18, 19]. The relative signs in (15)–(16) suggest a *non-abelian version of the twisted boundary algebra*. The notation $(q \pm \varepsilon h^{(i)})$ in (23) is a book-keeping device to characterize the relative signs of the dynamical shifts of the deformation parameter $q$ along an auxiliary space $i$. In the abelian case, they acquire an actual meaning as an additive shift by a Cartan algebra element.

In the non-abelian case however, a technical problem arises in formulating exactly the shifts. It was solved in [13] by introducing a star-product $*$ entailing a formal, ordered, Taylor series expansion, both to define $R * T * T$ and $R(q`` + "h)$ (where "$+$" denotes the operation of non-abelian shift). The dynamical deformation is implemented by a Drinfel'd twist as in the abelian case [20,21], verifying a twisted coboundary equation expressed in terms of the star-product. However, it is not easy to manipulate explicitly such objects when we will need to define coproduct, comodule or trace structures, and a reformulation of these equations should be required. In addition, the notion of a non-abelian dynamical quantum reflection algebra is yet undefined strictly speaking, and its structure and properties need to be further described.

Finally, a deeper issue arises in the quantization procedure of (15)–(16). The presence of an associative $\Omega_{123}$, which transforms the classical YBE into a *modified* classical YBE for the matrix $a$, and adjoint classical YBE into a *modified* adjoint classical YBE for the matrices $(a, s)$, together with the non-equivariance of $a$, indicates an obstruction to associativity in the quantum case. Indeed, $\Omega_{123}$ may be obtained as a classical limit of the quasi-associator $\Phi_{123}$ in the quantum Yang–Baxter associator equation [22]:

$$R_{12}\,\Phi_{312}\,R_{13}\,\Phi_{132}^{-1}\,R_{23}\,\Phi_{123} = \Phi_{321}\,R_{23}\,\Phi_{231}^{-1}\,R_{13}\,\Phi_{213}\,R_{12}\,. \tag{24}$$

Without the dynamical deformation, when $\Phi_{123} = \mathbb{I}_{123} + \hbar^2\,\Omega_{123} + o(\hbar^2)$, one derives the known (non-dynamical) modified classical YBE for an $a$ matrix from (24). A Drinfel'd (non-abelian) twist of the structure (24) should yield a dynamical deformation with (15) as its classical limit. In addition, $\Omega_{123}$ obeys the coproduct relations for $\Phi_{123}$ at first leading order, hence realizes indeed a consistent first non-trivial order for a quantum associator $\Phi_{123}$.

Non-equivariance of the matrix $a$ does not play any role at the level of classical Yang–Baxter equations, since it yields a term of higher order in $\hbar$. It prevents however such operations as $R_{ij}\,K_n(q`` + "h_i`` + "h_j) = K_n R_{ij}$, required in the establishing of the quantum YBE as a consequence of associativity of the algebra.

We conjecture here that the two obstructions may in fact combine to yield a quasi-associative dynamical reflection algebra, and contribute consistently to the generalized quantum dynamical YBE for such an algebra. Otherwise they may alternatively characterize an obstruction to quantization, to be lifted precisely by the additional term to the CSG Lagrangian identified in [7] to reestablish integrability. Only an explicit construction of the relevant objects will allow to clarify this issue. We shall leave this particular discussion on associativity issues in the quantization of (15)–(16) as an open issue.

The quantization of (15)–(16) must thus deal with three new features for a reflection algebra: (i) quasi-coassociativity; (ii) non-abelian dynamics; (iii) $\mathbb{Z}_2$-twist to get twisted boundary dynamical algebra. Since the last two features can be implemented at least formally by known procedures over a general quasi-Hopf algebra [20,22], the quantization of (15)–(16) should be undertaken as follows:

1. Construct a quasi-associative quasi-Hopf algebra with (24) as its quantum Yang–Baxter equation, and define explicitly coproducts and traces by characterizing a consistent quantum coassociator $\Phi_{123}$.

2. Implement the non-abelian dynamical shifts by an explicit Drinfeld twist procedure on the lines of [20,21], extended to the non-abelian case as formulated in [13].

3. Implement the $\mathbb{Z}_2$-twist to construct a dynamical *reflection* algebra structure of twisted boundary algebra type on the lines of [14].

The final result should consistently yield (15)–(16) as a classical limit. Should any obstruction arise, it will point out at the core of the issues addressed in the introduction regarding quantum CSG integrability. Note that one may independently implement step 1, then step 3, to obtain the (a priori not yet known) quantum version of the *modified* CYB-reflection algebra, without the added complication of dynamical deformation. It would be interesting if explicit examples were then to be identified.

We wish to point out that a "mirror" quantization was proposed in [23,24], in the sense that the authors exploit the identification of CSG as a gauged $SU(2)/U(1)$ WNZW with deformation by first quantizing it, before gauging it to yield CSG. The initial $(a,s)$ matrix structure with a modified classical Yang–Baxter Poisson algebra is already present, albeit non-dynamical. We expect that the gauging procedure after quantization of the Poisson bracket structure, generates the non-abelian dynamical deformation of the $(a,s)$ pair identified here, on lines similar to the construction in [25] of dynamical Calogero–Moser $r$-matrices from symmetric spaces $(a,s)$ structure, after Hamiltonian reduction.

We are not aware of a discrete version of the complex sine-Gordon model, which would help in its quantization. The dynamical nature of the $(a,s)$ matrix structure seems to indicate that any such discrete system would look more like a spin-Calogero–Moser or spin-Ruijsenaars–Schneider model. Explicit uses of the integrability structures (such as Bethe Ansatz constructions) for integrable quantum discrete systems with underlying dynamical quantum algebra can be found in e.g. [26]. However, let us note that the dynamical deformation of these structures is carried by an abelian algebra, whereas in our case it is a non-abelian algebra which is at stake.

The last question to address regarding a quantization of the classical $a,s$ Poisson structure derived in our paper, is related to the key feature of this classical algebra as a dynamically deformed structure parametrized by the field variables $\psi$, $\bar{\psi}$, identified as coordinates on the target manifold $SU(2)/U(1)$ of the related WZWN model. For each value of the space variable $x$, a deformation algebra $\mathcal{U}_{\psi,\bar{\psi}}(su(2))$ is thus to be defined. The classical $a,s$ Yang–Baxter structure is ultralocal (contrary to the $L$-Poisson structure which is not ultralocal, an issue which we will also address in this respect).

The monodromy matrix for the Lax matrix at the classical level, derived by Maillet in 1985 [8], is naturally defined as following from an evolution along the $x$-axis. In all known cases when a quantum monodromy matrix is defined for a quantum integrable theory, the monodromy structure is simply obtained from the iteration of the coproduct (for an RTT structure) or a comodule (for an RKRK structure) of the underlying quantum algebra, and is identified as a tensor product over the "neighboring" quantum spaces. In this case, by contrast, the monodromy is taken over the index $x$, i.e. over the index of *deformation* manifold of the algebra, not of *quantum spaces* or algebra itself. As far as we know, such a "monodromy matrix" as tensor product of algebras with distinct but isomorphic deformation moduli spaces, has never been defined.

We expect that the resolution of this problem will require the introduction of intertwining structures "between" the different $x$-labeled deformed algebras, which should then generate the Lax matrix, and the terms $\partial_x a\, \delta(x-y)$ and $(s+s)(\partial_x - \partial_y)\delta(x-y)$, by a classical limit. More precisely, the Lax matrix may arise as a combination of a $K$-matrix in a reflection algebra structure at fixed $x$, ultralocal, yielding the $\delta(x-y)$ commutator terms of the Poisson bracket, with matrices $A$ and/or $S$, yielding the $\partial_x a\, \delta(x-y)$ and $(s+s)(\partial_x - \partial_y)\delta(x-y)$ terms.

The reflection algebra structure identified here is therefore completely independent of any structure underlying defect/boundary CSG considered in [27, 28]. Defect and boundaries in this last case occur in continuous space at a fixed value $x_0$. By contrast, "reflection" in our CSG structure occurs in a different, two-dimensional, space at every point $x$, and depends on the fields $\psi, \bar{\psi}, \pi, \bar{\pi}$ at this point through the quantum Lax matrix $L$. This may lead to a construction of potentially interesting new integrable models, coupling spin-like variables $s_i(x)$ with the fields $\psi, \bar{\psi}, \pi, \bar{\pi}$, in the same line as in the construction of [29].

In conclusion, new quantum structures need to be defined precisely at each step of our conjectural procedure. They are interesting for their own sake and should be explored in detail. If the procedure can be pursued up to the end, it will provide a full quantization of CSG as an integrable theory, after modifications including the anomalies mentioned in the

beginning of this section.

# Acknowledgements

Work partially sponsored by CNRS. J.A. wishes to thank LAPTh Annecy for their kind hospitality.

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
