# Peer review of "Algebraic structure of classical integrability for complex sine-Gordon"

_SciPost Physics, doi:SciPost Phys. 8, 033 (2020)_

## Round 1 · Referee Report · Anonymous (Referee 1) · 2019-12-20

Strengths

  1. Comprehensive analysis of the complex sine-Gordon model in the Hamiltonian formalism.
  2. Uncovering of the novel Poisson bracket structure of the model.
  3. Interesting discussion of how the classical structure could be quantized.

Weaknesses

  1. Not clear whether this approach will lead to a successful quantization of the model, are there many similar model where that is known (e.g. sine Gordon theory is related to a spin chain, can we expect the same for the complex sine Gordon?)

Report

The paper analyses the classical integrability of the complex sine-gordon theory in the Hamiltonian formalism as a precursor to a quantum treatment. There is a preliminary discussion of the quantum version of the model. The structure uncovered in rather non-standard and this makes the investigation interesting and novel and worthy of publication.

Requested changes

  1. bookkeeping -> book-keeping
  2. page 2 "We answer this last question in this paper". What question is this referring to?
  3. generally the grammar used in the paper is awkward. There are instances where an article should be used (the or a), for example. The use (or lack) of commas is another issue. I won't go through the paper line by line pointing out the language deficiencies, but suggest a careful edit to make it read better.

---

## Round 2 · List of Changes

Dear Editor,

We have revised our manuscript according to the requested changes of the referee. Here are the major modifications:
- page 2, the sentence has been modified to clarify the point.
- we added three paragraphs in section 3 :
a/ page 8, a comment on an alternative procedure of quantizing the complex sine-Gordon model (with refs 23 and 24 added);
b/ page 8, a comment on the potential existence of discrete quantum systems related to CSG (with ref. 26).
c/ page 9, a final paragraph about the feasibility of the quantization itself.

In addition, the grammar and the punctuation have been revised.
Best regards,
The authors

---

## Editorial Decision

published